# Evaluation of Population–Food Relationship from the Perspective of Climate Productivity Potential: A Case Study of Eastern Gansu in Northwest China

**Junqi Cheng and Shuyan Yin ***

School of Geography and Tourism, Shaanxi Normal University, Xi'an 710119, China; chengjq789@163.com
* Correspondence: yinshy@snnu.edu.cn

**Abstract:** Suffering from the double blow of the new crown pneumonia epidemic and floods, food security issues have once again become a source of concern. Eastern Gansu is an important dry farming area in northwestern China, and agricultural production has been greatly affected by climate change. Based on the climate data of 17 national meteorological stations in eastern Gansu from 1961 to 2020 and the data on population, grain planting area and grain production in each region from 1986 to 2019, using the Thornthwaite Memorial model, this paper analyzed the climate production potential ($T_{SPV}$), population carrying capacity and population carrying capacity index in eastern Gansu, and then revealed the relationship between population and food in eastern Gansu. The findings and results revealed that: (1) over the past 60 years, the temperature in eastern Gansu has been increasing and precipitation has been decreasing; (2) $T_{SPV}$ has been increasing. Moreover, the spatial distribution was significantly different, showing a trend of decreasing from the southeast to the northwest. Lintao, Huining, and Jingtai displayed a decreasing trend, while other areas exhibited an increasing trend. Precipitation was the main limiting factor for $T_{SPV}$; (3) Grain production continued to increase due to changing hydrothermal conditions and improved production efficiency. Cultivated land–population carrying capacity and climate production potential–population carrying capacity ($T_{SPV}$–population carrying capacity) both exhibited a significant increasing trend ($p < 0.01$). Cultivated land–population capacity increased from southeast to northwest, and all stations expressed an increasing trend. $T_{SPV}$–population carrying capacity increased from southeast to northwest, and the whole region displayed an increasing trend. Even in extremely reduced production years, $T_{SPV}$–population carrying capacity was also greater than cultivated land–population carrying capacity. This revealed that, under ideal conditions, $T_{SPV}$–population carrying capacity can fully meet the needs of the current population. (4) The population carrying capacity index showed a significant downward trend ($p < 0.01$). It showed a trend of decreasing from south to north, and whole area underwent a decreasing trend consistently, indicating that the population–food relationship in eastern Gansu tended to be balanced. This result was conducive to correct assessment of the relationship between people and food in the study area, and provided a reference for formulating food policies.

**Keywords:** cultivated land; population carrying capacity; climate production potential

## 1. Introduction

Food security is related to national security and social stability, and is an important guarantor of sustainable economic development [1]. For a long time, the production structure, trade circulation, supply and demand relationship, etc. have been sources of great concern, both in China and abroad [2–9]. Food supply, water shortage, food security and climate change are the biggest challenges facing mankind in the future [10,11]. These challenges are manifested in the relationship between food supply and climate change, climate change and water shortage and water shortage and food supply [12–14].

Climatic production potential refers to the highest plant biological yield or economic yield that can be obtained from a unit area of land when the conditions of soil, nutrients

and other conditions are in their most suitable states and climate resources such as water and temperature are fully and rationally used [15,16].

Commonly used climate production potential estimation models include the Miami model [17], the Thornthwaite Memorial model [18], the Chikugo model [19] and the integrated vegetation model [20]. In recent years, domestic and foreign scholars have achieved rich results in the research of climate production potential [21–25].

Since 2021, affected by the impact of the new crown pneumonia epidemic and floods, concerns over food security have seen a new upsurge [26]. Gansu is a large agricultural province in China and is sensitive to climate change [27]. To date, research on Gansu Province has mainly focused on the changing characteristics of production potential [28,29], while the population–food relationship, based on climate production potential, has received little attention. Most of the eastern part of Gansu is located in the Loess Plateau. Serious soil erosion, frequent natural disasters, limited arable land resources, and tense population–food relations are features of the area. Therefore, it was particularly important to study the population–food relationship in eastern Gansu. This paper was based on the current situation regarding population and cultivated land in eastern Gansu, and based on the theory of the relationship between humans and food. Using the climate productive potential model to study the population–food relationship in eastern Gansu will be of great significance to efforts to adjust and optimize agricultural planting structures, improve agricultural productivity, and actively respond to climate change.

This study aimed to evaluate the population–food relationship in eastern Gansu based on the $T_{SPV}$ and the main line of population–food. The overall objectives of this study were: (1) characteristics of climate change in eastern Gansu; (2) temporal and spatial change characteristics of $T_{SPV}$; (3) the changing characteristics of cultivated land–population carrying capacity and $T_{SPV}$–population carrying capacity and the relationship between them; (4) to analyze the changing characteristics of the population carrying capacity index, and then reveal the change characteristics of the population–food relationship. This research could provide a scientific basis for the national population policy and food production.

## 2. Materials and Methods

### 2.1. Study Area

The eastern part of Gansu is located in northwestern China (Figure 1), on the upper reaches of the Yellow River. Except for the alpine climate in the Gannan Plateau, the eastern part of Gansu belongs to the monsoon climate. From south to north, it can be divided into 6 climate zones, namely: northern subtropical humid zone, warm temperate humid zone, alpine zone, cold temperate semi-cold zone, humid area, cold temperate semi-arid area. The distribution of precipitation is more in the south than in the north, with a large difference between the two and average annual precipitation ranging from 200 to 850 mm [30]. The area is rich in landform types and complex topographically, forming three natural areas of the Longzhong Loess Plateau, Gannan Plateau and Longnan Mountains [31]—which are semi-arid and semi-humid rain-fed agricultural areas. The main food crops include wheat, corn, potatoes and beans. Wheat is dominant, accounting for more than half of all crops. The main economic crops are rape, beet, and sunflower. The population is mainly distributed in the Longzhong Plateau and the Longdong area, mainly engaged in agricultural production.

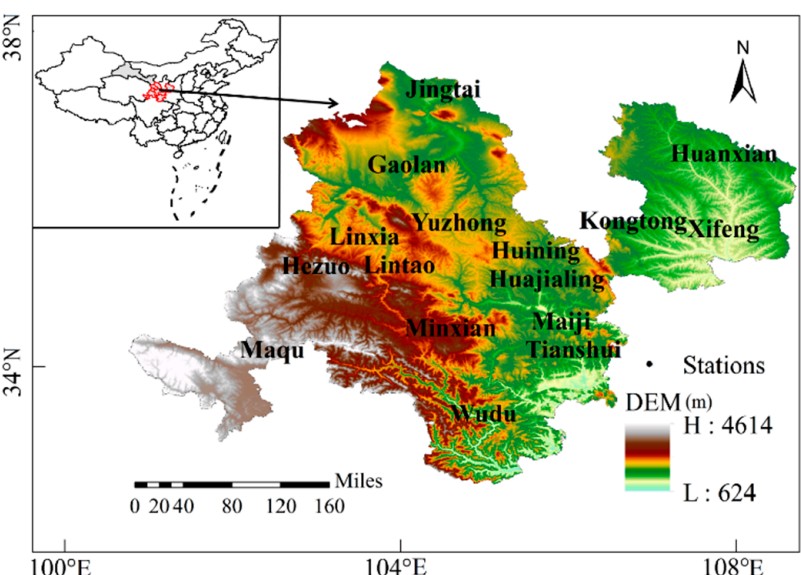

**Figure 1.** Location of the study area.

*2.2. Data Sources*

After comprehensively considering the completeness and continuity of the data and strict screening, the annual average temperature and precipitation data of 17 national meteorological stations in eastern Gansu were chosen, ranging from 1961 to 2020. The data came from China National Meteorological Data Center (http://data.cma.cn/) (accessed on 15 September 2021). The data of selected stations were strictly checked and controlled, including the elimination of outliers and error values, time consistency inspection, determining whether the daily minimum temperature was greater than the maximum temperature, extreme value inspection, etc. [32]. The total population, grain output and grain planting area of each region came from the *Gansu Statistical Yearbook*, and the time series was 1986–2019.

*2.3. Methods*

2.3.1. Evaluation Model of Climate Production Potential and Population–Food Relationship

(1) Climate production potential

The Miami model can reflect the influence of a single factor of water and heat on potential productivity under natural conditions [33]. The Thornthwaite Memorial model considers the evapotranspiration, closely related to plant yield, on the basis of the Miami model, reflecting the comprehensive influence of multiple meteorological elements [25]. Because these two models are relatively simple and require few parameters, they are widely used in the study of large-scale climate production potential change patterns [34]. Therefore, this study used the Thornthwaite Memorial model to calculate climate production potential, and the calculation formula was as follows [35]:

$$T_{SPV} = 3000[1 - e^{-0.0009695(V-20)}] \tag{1}$$

$$V = 1.05R \times [1 + (1.05R/L)^2]^{-0.5} \tag{2}$$

$$L = 300 + 25T + 0.05T^3 \tag{3}$$

where $V$ is the annual average actual evapotranspiration (mm); $L$ is the maximum annual evapotranspiration (mm), an empirical function representing the annual average temperature; $T$ is the annual average temperature (°C); and $R$ is the annual precipitation (mm). If and only if $R > 0.316 L$, we calculated according to Formulas (1) and (2); If $R \leq 0.316L$, then $V = R$.

Climate Carrying Capacity Index

The ratio of the actual population to $T_{SPV}$–population carrying capacity was defined as the climate carrying capacity index ($C_b$). The smaller the value, the more the food surplus, the more coordinated the population–food relationship [22].

$$C_b = \frac{P_a}{kA_f \bullet C_{bf}}$$
(4)

where $P_a$ is the actual population (person) in the assessment year; k is the climate–grain conversion index, expressed as the proportion of grain output per unit area in $T_{SPV}$, which characterizes the ability of $T_{SPV}$ to convert into food. When k is 1, $C_b$ represents the maximum climate carrying capacity level under ideal conditions—that is, the population–food relationship state; $A_f$ is the estimated annual grain planting area ($hm^2$); $C_{bf}$ is $T_{SPV}$–population carrying capacity (person·$hm^{-2}$), which represents the number of people that can be supported by the $T_{SPV}$ per unit area at a certain productivity level [36], namely:

$$C_{bf} = \frac{T_{SPV}}{N_f}$$
(5)

where $T_{SPV}$ is the climate production potential (kg·$hm^{-2}$ ·$a^{-1}$), and $N_f$ is the per capita food demand standard for different levels (kg person$^{-1}a^{-1}$). If k in Equation (4) is 1, and $T_{SPV}$ in Equation (5) is the actual grain yield (kg·$hm^{-2}$), then $C_{bf}$ represents the number of people that can be carried by the grain output per unit area of cultivated land—that is, cultivated land–population carrying capacity ($C_{lf}$). $C_b$ can represent the current population carrying capacity of cultivated land and grain production—that is, the cultivated land carrying capacity index ($C_l$), which can describe the current population–food relationship status of actual food supply. Relevant domestic scholars have calculated, based on China's national conditions and the per capita nutritional calorific value standards published by the Food and Agriculture Organization of the United Nations, that China's per capita food consumption of 400 kg/a met the requirements of nutritional safety [37]. Therefore, this article regarded it as the food consumption standard in eastern Gansu.

Grading evaluation standard of bearing capacity index

According to the value of $C_b$, the climate carrying capacity of eastern Gansu was divided into three categories: food surplus, population–food balance, and population overload, totaling eight levels [38,39] (Table 1).

**Table 1.** Evaluation standard of carrying capacity index.

| Population Carrying Capacity | | Standard |
|---|---|---|
| Type | Level | |
| Food surplus | More than affluence | C < 0.5 |
| | Affluence | 0.5 ≤ C ≤ 0.75 |
| | Surplus | 0.75 < C ≤ 0.875 |
| Population–food balance | Balance and surplus | 0.875 < C ≤ 1.0 |
| | Critical overload | 1.0 < C ≤ 1.125 |
| Population overload | Overload | 1.125 < C ≤ 1.25 |
| | Obviously overload | 1.25 < C ≤ 1.5 |
| | Serious overload | C > 1.5 |

### 2.3.2. Trend Analysis

The trend line analysis method mainly uses the least square method to simulate the change trend of $T_{SPV}$ over time per pixel [28], which can eliminate the influence of extreme

climate in individual years on $T_{SPV}$ to a certain extent [40]. Its calculation formula is as follows:

$$\theta_{slope} = \frac{n \times \sum\limits_{i=1}^{n} i \times Y_i - \sum\limits_{i=1}^{n} i \sum\limits_{i=1}^{n} Y_i}{n \times \sum\limits_{i=1}^{n} i^2 - \left(\sum\limits_{i=1}^{n} i\right)^2} \tag{6}$$

where $\theta_{slope}$ is the trend slope; $Y_i$ is the standard $T_{SPV}$ of the *i*-th year (kg·hm$^{-2}$·a$^{-1}$); and *i* is the annual variable. When $\theta_{slope} < 0$, it indicates that $T_{SPV}$ is decreasing. On the contrary, there was an increasing trend observed in this study.

### 2.3.3. Mutation Test

The Mann–Kendall (MK) mutation test method is a time series trend analysis method. It is recommended by the World Meteorological Organization and widely used in non-parametric statistical test methods. It has the advantage that it does not need to follow a certain distribution and is not disturbed by outliers [41]. Therefore, the M–K mutation test method was used herein to test the change trend of temperature, precipitation and $T_{SPV}$. The specific method can be found in the literature [41].

## 3. Results

### 3.1. Characteristics of Temperature and Precipitation Changes

Figure 2 illustrates the changes in annual average temperature and precipitation in eastern Gansu. The annual average temperature (Figure 2a) increased at a rate of 0.3 °C/10a, a significant increase trend ($p < 0.01$). The multi-year average temperature was 7.83 °C. The average temperature was highest in 2016 at 9.16 °C, which was 1.33 °C higher than the average. The average temperature was lowest in 1967 (6.47 °C). The multi-year average precipitation (Figure 2b) decreased at a rate of 1.3 mm/10a, with a multi-year average of 455.9 mm, the largest in 1967 (628.66 mm), and the smallest in 1997 (332.2 mm). The inter-annual difference was significant, demonstrating that the climate in eastern Gansu has shown a trend of warming and drying.

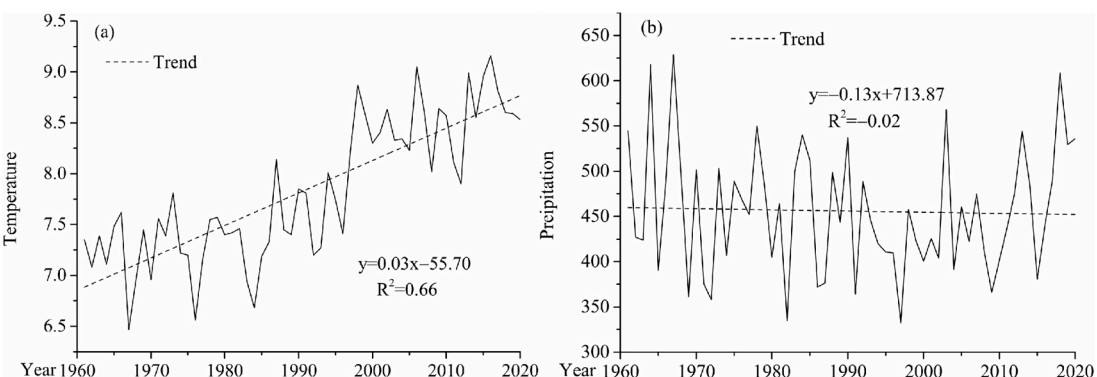

**Figure 2.** Annual variations of mean temperature (**a**) and precipitation (**b**) in east Gansu from 1961 to 2020.

### 3.2. Characteristics of Climate Production Potential Changes

The average $T_{SPV}$ of eastern Gansu from 1961 to 2020 was 8092.4 kg·hm$^{-2}$·a$^{-1}$, and the average annual $T_{SPV}$ of each region was 6899.02–9809.1 kg·hm$^{-2}$·a$^{-1}$. The overall $T_{SPV}$ increased at a rate of 93.2 kg·hm$^{-2}$ (Figure 3a). There were two obvious stages of change, which displayed a decreasing trend from 1961 to 1996. The trend coefficient was −105.7 kg·hm$^{-2}$·10a$^{-1}$, and the minimum value appeared in 1971. There was an increasing trend from 1997 to 2020, with a coefficient of 342.2 kg·hm$^{-2}$·10a$^{-1}$. The maximum value appeared in 2013.

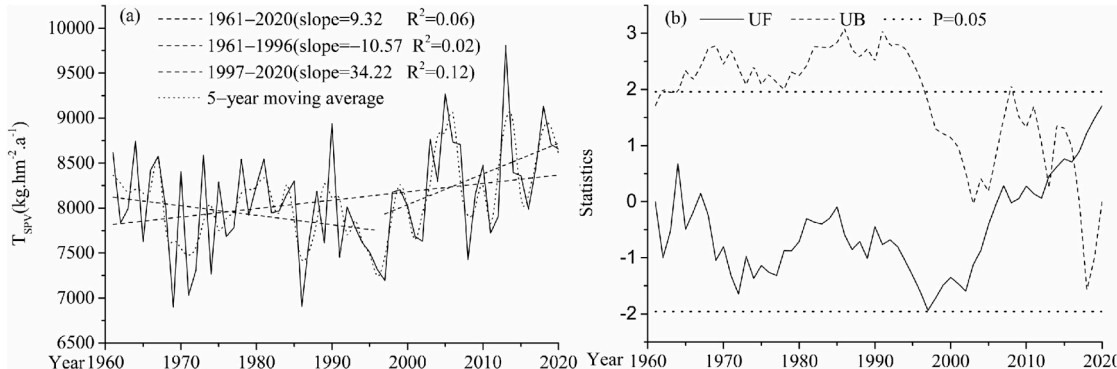

**Figure 3.** Change trend of $T_{SPV}$ (**a**) and Mann-Kendall mutation test (**b**) in East Gansu from 1961 to 2020.

MK mutation analysis revealed (Figure 3b) that UF was less than 0 from 1961 to 2008 (except 1964); thus, $T_{SPV}$ has been decreasing. Since 2009, it has been greater than 0 and continued to increase, indicating that $T_{SPV}$ has begun to increase. The UF and UB curves intersected in the confidence interval, but there was no mutation.

From the perspective of interdecadal changes (Table 2), the average value of $T_{SPV}$ in eastern Gansu was the largest during 2011–2020, and was the smallest in the 1970s.

**Table 2.** Interdecadal change of $T_{SPV}$.

| Decade | 1961–1970 | 1971–1980 | 1981–1990 | 1991–2000 | 2001–2010 | 2011–2020 |
|---|---|---|---|---|---|---|
| Tspv | 8294.05 | 7993.59 | 8422.86 | 8257.20 | 8794.44 | 8903.81 |

The spatial difference of $T_{SPV}$ was significant, decreasing from southeast to northwest (Figure 4a). The maximum $T_{SPV}$ was greater than 9000 kg·hm$^{-2}$·a$^{-1}$ in Tianshui and Longnan, and greater than 9800 kg·hm$^{-2}$·a$^{-1}$ in Wudu. The $T_{SPV}$ of Gaolan was 5022.32 kg·hm$^{-2}$·a$^{-1}$, which was less than 6000 kg·hm$^{-2}$·a$^{-1}$ in northwest China. The change was not significant in most areas (Figure 4b), with an increasing trend accounting for 82.35%. The increase was most significant in Jingyuan, and only Lintao, Huining and Jingtai showed a decreasing trend.

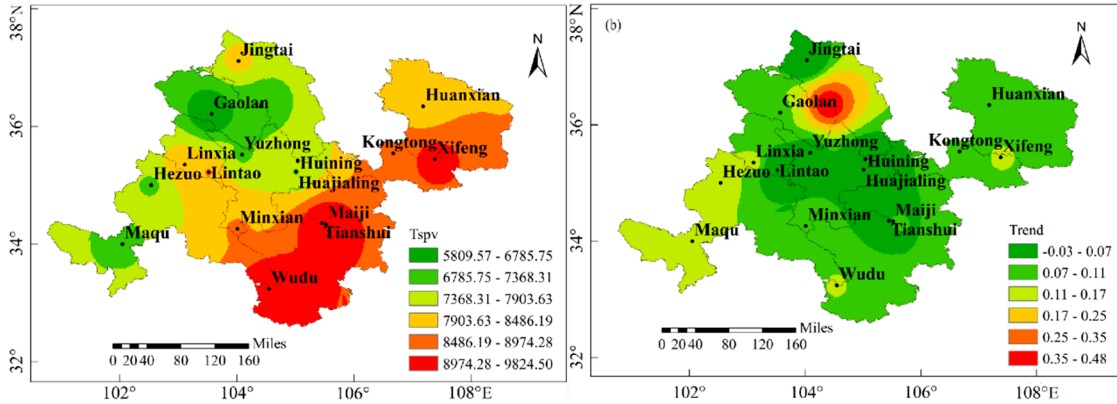

**Figure 4.** Spatial distribution of average value (**a**) and linear changing trend (**b**) of $T_{SPV}$ in East Gansu from 1961 to 2020.

In order to further study the impacts of temperature and precipitation on $T_{SPV}$, a correlation analysis was carried out between temperature and precipitation at each meteorological station and $T_{SPV}$ (Table 3). It was found that the correlation coefficient between

temperature and $T_{SPV}$ was −0.13–0.835, with significant regional differences. Only 29% of stations passed the significance test. The correlation coefficient between rainfall and $T_{SPV}$ ranged from 0.295 to 0.975, showing consistent positive correlation, and all passed the significance test. Therefore, precipitation was the main limiting factor of $T_{SPV}$ in eastern Gansu province.

**Table 3.** The correlation between $T_{SPV}$ and temperature and precipitation.

|  | Temperature | Precipitation |
|---|---|---|
| Gaolan | −0.028 | 0.434 ** |
| Hezuo | 0.634 ** | 0.791 ** |
| Huajialing | 0.158 | 0.889 ** |
| Huanxian | −0.07 | 0.956 ** |
| Huining | −0.114 | 0.921 ** |
| Jingtai | 0.335 ** | 0.745 ** |
| Jingyuan | 0.299 * | 0.295 * |
| Kongtong | 0.1 | 0.959 ** |
| Lintao | −0.134 | 0.947 ** |
| Linxia | 0.169 | 0.945 ** |
| Maqu | 0.835 ** | 0.617 ** |
| Maiji | 0.071 | 0.939 ** |
| Minxian | 0.36 ** | 0.895 ** |
| Tianshui | −0.131 | 0.965 ** |
| Wudu | 0.051 | 0.471 ** |
| Xifeng | 0.201 | 0.916 ** |
| Yuzhong | −0.103 | 0.975 ** |

Note: * represents 0.05 significance, ** represents 0.01 significance.

### 3.3. Variation Characteristics of Population Carrying Capacity

Figure 5 illustrates the temporal change characteristics of the population carrying capacity index in eastern Gansu over the past 34 years. The cultivated land–population carrying capacity and the $T_{SPV}$–population carrying capacity both showed a very significant increasing trend ($p < 0.01$). The interannual variation (Table 4) showed that the mean value of cultivated–population carrying capacity was largest in 2011–2019 and smallest in the 1980s, while the mean value of $T_{SPV}$–population carrying capacity was largest in 2011–2019 and smallest in the 1990s.

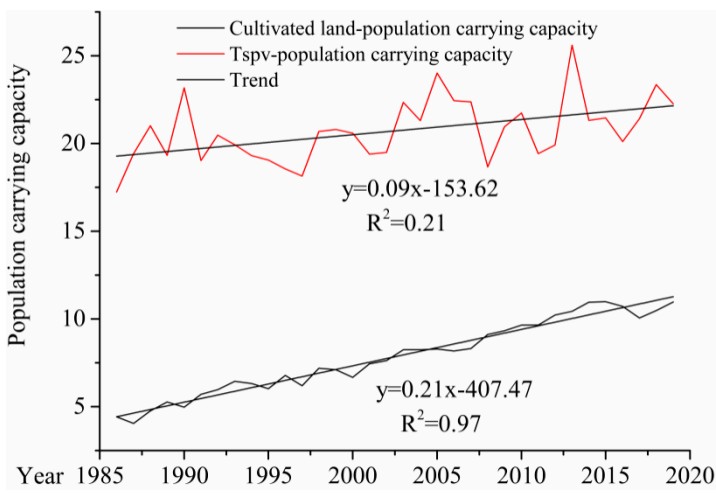

**Figure 5.** Change trend of population carrying capacity in East Gansu in 1961−2020.

**Table 4.** Decadal average value of mean annual population carrying capacity.

|  | 1986–1990 | 1991–2000 | 2001–2010 | 2011–2019 |
|---|---|---|---|---|
| Cultivated land–population carrying capacity | 4.70 | 6.44 | 8.44 | 10.50 |
| $T_{SPV}$—population carrying capacity | 20.03 | 19.66 | 21.28 | 21.65 |

By comparison, $T_{SPV}$–population carrying capacity was much larger than that of the cultivated land–population, and the cultivated land–population carrying capacity was 38% of $T_{SPV}$–population carrying capacity. The results indicated that the current $T_{SPV}$–population carrying capacity was larger than the actual demand, and the cultivated land–population carrying capacity had more space to use than $T_{SPV}$–population carrying capacity. This was mainly attributable to the improvement of land farming technology and the improvement of production efficiency. Therefore, in terms of the current population size and food demand, what affected the population–food relationship in eastern Gansu was the utilization rate of climate production potential and food production. Under the circumstance of increasing food demand, it is important to increase food production so that the increasing food supply can fully meet the demand.

From the perspective of spatial distribution and change trend, the cultivated land–population carrying capacity increased from southeast to northwest (Figure 6a). The minimum value was located in the eastern Huanxian (4.02 person·hm$^2$) and the maximum value was 20.01 person·hm$^2$, in Linxia. There was a large gap between the maximum value and the minimum value, so the spatial difference of cultivated land–population carrying capacity was significant. All stations showed an increasing trend (Figure 6b), with a variation coefficient of 0.4–4.2/10a. Linxia had the largest increase and Hezuo had the smallest increase. $T_{SPV}$–population carrying capacity (Figure 6c) decreased from southeast to northwest, with the maximum value reaching 24.75 person·hm$^2$ in Wudu, and the minimum value reaching 14.69 person·hm$^2$ in Gaolan. $T_{SPV}$–population carrying capacity also showed an increasing trend (Figure 6d), with a variation coefficient of 0.2–3.2/10a. The increasing trend decreased from northwest to southwest, among which Xifeng and Minxian were tested for significance and showed the most significant increase.

The annual mean of $T_{SPV}$ represented annual value of normal years; a 5% guarantee rate represents annual value of disaster, a 20% guarantee rate represents the annual value of yield reduction, an 80% guarantee rate represents the annual value of yield increase, a 95% guarantee rate represents the annual value of high yield, the minimum value represents the annual value of extreme yield reduction, and the maximum value represents the annual value of extreme yield increase (Table 5). It was observed that the normal annual value was closest to the increased annual value, indicating that $T_{SPV}$ in eastern Gansu was higher in most years. There was a large gap between extreme yield reduction and increase, and the minimum value was only about 71% of the maximum value, indicating significant interannual change caused by climate. From the perspective of population carrying capacity, under the normal annual value, the population carrying capacity was 21.11 person·hm$^2$, the yield decrease years were 19.82 person·hm$^2$, and the yield increase years were 22.04 person·hm$^2$. According to the ideal state, k value was set to 1, and the sown area of grain crops from 1986 to 2019 was used as the supply level of cultivated land. Even in the years with extreme production reduction, the $T_{SPV}$–population carrying capacity was larger than the cultivated land–population carrying capacity, indicating that the climatic carrying capacity level could fully meet the current population needs under the ideal state.

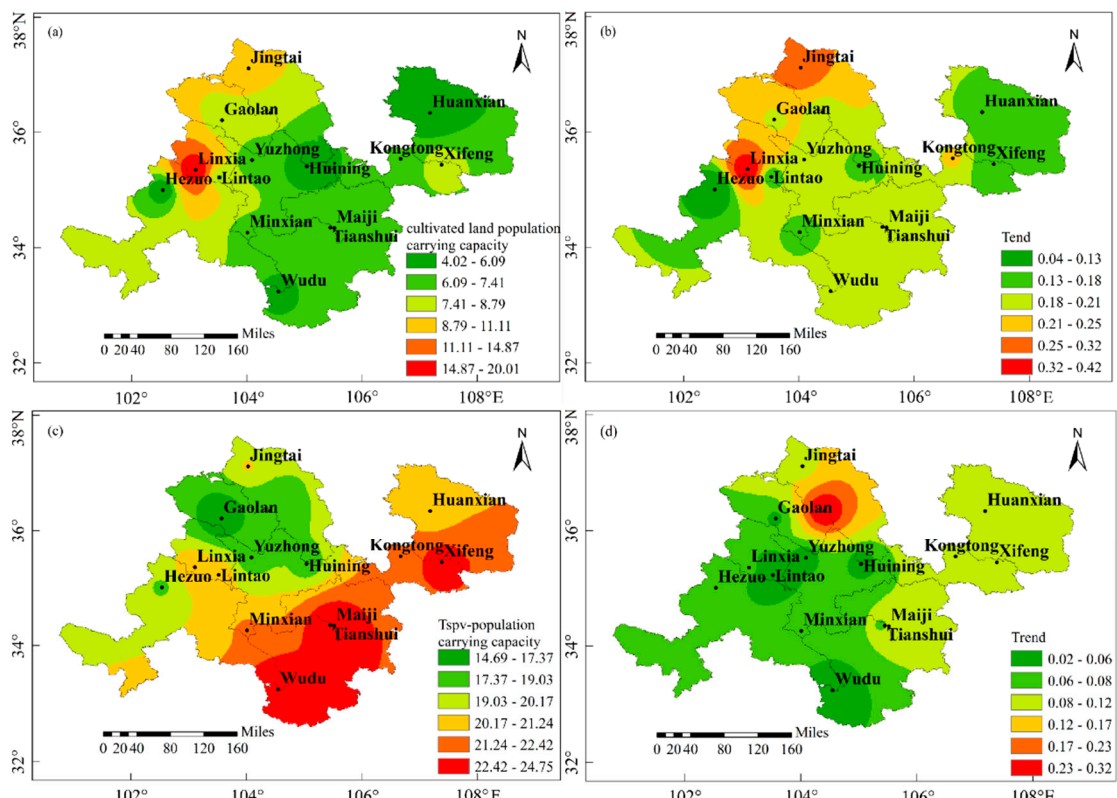

**Figure 6.** Spatial distribution of cultivated land–population carrying capacity (**a**) and linear changing trend (**b**); $T_{SPV}$–population carrying capacity (**c**) and linear changing trend (**d**) in East Gansu from 1961 to 2020.

**Table 5.** Statistical characteristics of $T_{SPV}$–population carrying capacities under different food demand levels.

| | Multi-Year Average | 5% Guarantee Rate | 20% Guarantee Rate | 80% Guarantee Rate | 95% Guarantee Rate | Minimum | Maximum |
|---|---|---|---|---|---|---|---|
| $T_{SPV}$ | 8444.32 | 7377.16 | 7926.52 | 8816.87 | 9234.31 | 7164.45 | 9989.25 |
| Population carrying capacity | 21.11 | 18.44 | 19.82 | 22.04 | 23.09 | 17.91 | 24.97 |

### 3.4. Population–food Relationship Assessment

The population carrying capacity index of eastern Gansu (Figure 7) exhibited a significant downward trend from 1986 to 2019 ($p < 0.01$), indicating that the relationship between population and food gradually approached equilibrium from overloading. The population carrying capacity index fluctuated greatly between years, and only the period 2013–2016 was in a state of population–food balance. Conversely, the other years were in a state of population overload, among which 1989–1991 was the most serious.

In terms of spatial distribution, the population carrying capacity index in eastern Gansu displayed a decreasing trend from south to north (Figure 8a); that is, from south to north, the population–food relationship gradually changed from population overload to population and food in balance. The population overloading was the most serious in Minxian, where the population carrying capacity index reached 2.52, and the smallest in Jingtai at 0.74. The changes in population carrying index (Figure 8b) showed a decreasing trend in all places except Minxian, and the decrease was most significant in Linxia, which passed the 0.01 significance test. It was noted that the population carrying capacity index had significant spatial differences and decreased in the whole region.

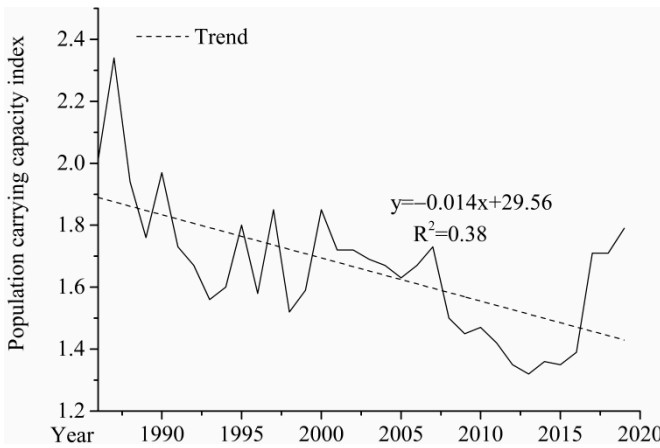

**Figure 7.** Change trend of population carrying capacity index in East Gansu from 1961 to 2020.

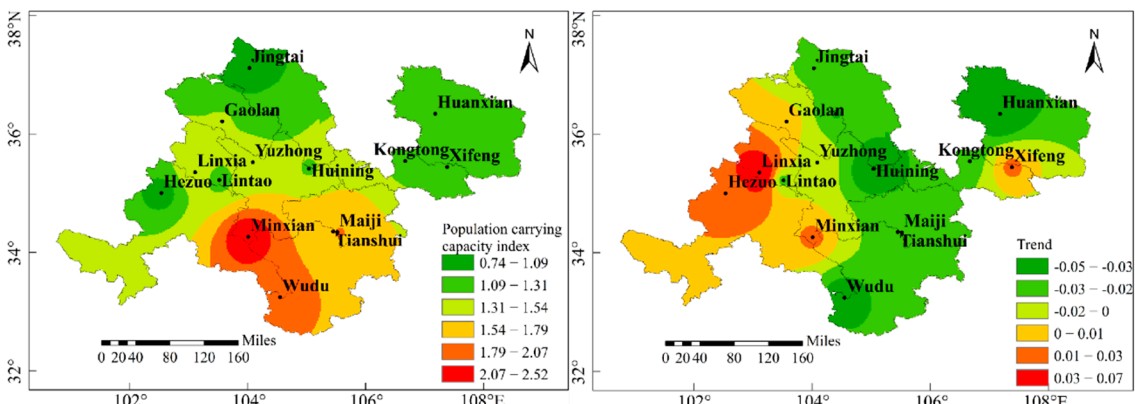

**Figure 8.** Spatial distribution of Population carrying capacity index (**a**) and linear changing trend (**b**) in East Gansu from 1961 to 2020.

## 4. Discussion

Based on the Thornthwaite Memorial model, this study explored the spatial and temporal distribution of climate productivity potential and population carrying capacity in eastern Gansu from 1961 to 2020. It was determined that the temperature in eastern Gansu increased significantly over the last 60 years, while the precipitation fluctuation decreased and the climate tended to be warm and dry. The research results on dryness and wetness in northwest China [33,42,43] all showed that the climate in northwest China was characterized by warm-dry and warm-wet alternation. However, the overall climate change could only be explained at a large scale, and interpretation of the laws of climate change in small regions is limited. More detailed research is needed to better reveal the laws of climate change.

The study found that the climatic productivity potential in eastern Gansu showed an increasing trend. Luo Yongzhong et al. [29] found that the temperature production potential in Gansu Province increased significantly, the precipitation production potential decreased slightly and the average climate production potential showed a gradually decreasing trend. The conclusions of this study may be related to the selected indicators, the choice of meteorological stations or the size of the study area.

The climatic productivity–population bearing capacity of eastern Gansu province was larger than that of cultivated land–population bearing capacity. Therefore, in terms of continuous population growth [44] and increasing food demand, it is better to increase land use efficiency, land quality and security to promote the sustainable development of agriculture and improve food security. To improve land use efficiency and quality, more environmentally friendly technologies and digitalization approaches [45–49] could be

applied. These include digital soil mapping and management strategies that could be used for accurate and efficient agricultural management (e.g., irrigation, fertility and carbon sequestration) and yield improvement. In addition, the eastern part of Gansu is located in the arid area of northwest China. Water resources are the main factor limiting agricultural development. We should vigorously develop water-saving agriculture, improve irrigation technology, cultivate new varieties, develop characteristic agriculture, strengthen education, improve residents' awareness of water-saving practices and promote the harmonious development of population and grain output.

The population carrying capacity index showed a downward trend, indicating that the population–food relationship has been gradually easing. This could be attributed to many factors; in terms of food policy, the state adopted a series of policies to benefit farmers, and investments in the construction of farmland water conservancy infrastructure have also increased; in terms of grain production prices, with the support of the national minimum purchase price policy, grain production prices have been stable. In the field of agricultural technology, mechanization has gradually covered, and improvement of production efficiency has mobilized farmers' enthusiasm for production, while saving significant labor for rural areas [50]. These have combined to increase the total grain output in eastern Gansu to varying degrees. In addition to satisfying self-sufficiency, grain production in some counties (districts) could also be exported. The population–food relationship has thus continued to maintain a stable and harmonious state.

This study was not without shortcomings. Although the selected stations covered every prefecture-level city in eastern Gansu, there were defects in the uneven distribution of meteorological stations. Data sources for population, grain yield and grain planting areas were sourced from *Gansu Statistical Yearbook*; however, since the Gannan region has been dominated by animal husbandry, the planting area of Maqu was missing and the time series of Hezuo data was short. In addition, with the continuing development of society and the progress of the times, some counties, such as Xifeng and Wudu, have become districts. As a result, the population and grain sown area changed greatly in a short period of time. All these deficiencies reduced the accuracy of our research. The Thornthwaite Memorial model is a climate statistical model obtained by fitting the empirical relationship between vegetation net productivity and climate factors. Many researchers have equated climate productivity potential with potential vegetation net primary productivity [51,52]. The model only considers the impacts of temperature, precipitation and other conditions on the climatic productivity potential. However, grain yield is also affected by other factors such as sunshine, radiation intensity and soil conditions. Further research on the model is needed to improve the accuracy of the research.

## 5. Conclusions

Over the past 60 years, the temperature in eastern Gansu has increased significantly, while precipitation has decreased, and the climate has tended to be warm and dry.

$T_{SPV}$ has been increasing, with a decreasing trend from southeast to northwest in space. Changes in most areas were not significant. Precipitation was the main constraint on climate productivity.

The cultivated land–population carrying capacity and $T_{SPV}$–population carrying capacity in eastern Gansu both exhibited a significant increase ($p < 0.01$). Cultivated land–population capacity increased from southeast to northwest, and all stations showed an increasing trend. $T_{SPV}$–population carrying capacity decreased from southeast to northwest, and also displayed a global increase trend. Even the extreme reduction in annual $T_{SPV}$–population carrying capacity was greater than the cultivated land–population carrying capacity, indicating that the ideal climate carrying capacity level was able to fully meet the current population needs.

The population-carrying capacity index showed a significant downward trend ($p < 0.01$). From south to north, the population–food relationship gradually changed from population

overload to population–food balance. The most serious population overload was in Minxian and the smallest was in Jingtai. All places displayed a decreasing trend except for Minxian.

**Author Contributions:** Conceptualization, J.C.; Formal analysis, J.C.; Investigation, S.Y.; Methodology, J.C.; Validation, J.C.; Visualization, S.Y.; Writing—original draft, J.C. All authors have read and agreed to the published version of the manuscript.

**Funding:** This research was funded by the National Natural Science Foundation of China, grant number '42071112, 41771110'.

**Institutional Review Board Statement:** Not applicable.

**Informed Consent Statement:** Not applicable.

**Data Availability Statement:** Not applicable.

**Conflicts of Interest:** The authors declare no conflict of interest.

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
