# Peer review of "Evaluation of Population–Food Relationship from the Perspective of Climate Productivity Potential: A Case Study of Eastern Gansu in Northwest China"

_atmosphere, doi:10.3390/atmos13020287_

Round 1

Reviewer 1 Report

The authors analyzed the climate production potential, population carrying capacity and population carrying capacity index in eastern Gansu, China and then revealed the relationship between population and food, using a climate production potential model. The paper is of interest to the readership of the journal and I suggest publication with minor revision.

Minor comments:

  • Equations appear out of text/font in pdf file
  • Same for tables and figures
  •  

Author Response

Thank you very much for your valuable suggestions on the article. We have revised the article, please take a look.
  Wish you all the best.

Reviewer 2 Report

In this paper, the relationships between population and food were analysed from an insight on climate productivity potentials using a Thornthwaite Memorial model. This study was based on a case in Gansu, China which is a large agricultural province. In brief, the submitted paper deals with an interesting topic which could provide valuable references for decision makers and policy makers to solve the increasing challenges of climate change and population on food security. The writing is clear and consistent, and the methodology is reasonable. However, in my view it lacks a thoughtful introduction which highlight the research problems and reviewed relevant literature in the research topic. Moreover, it also lacks an in-deep and critical discussion of the results. Therefore, a major revision is needed for further consideration for publication by atmosphere.

The paper should be deeply revised according to the following suggestions/comments and the detailed comments in the attached PDF file:

Abstract

Line 21 What is the reason for this trend is needed to be explained to give decision makers or policy makers indications.

Line 27-29 What are the implications of this study and the study conclusions is required.

Introduction

Line 38: It is better to use food security here.

Line 52: “… climate change” References?

In general, the research problems have not been fully analyzed in the Introduction. Why it is important to study the population-food relationships and why this is especially important for Gansu province? I recommend including more literature to explain these questions in the Introduction.

Materials and method

2.1

Line 70: References? Many similar statements in this study need references to support the ideas.

Line 75 Figure1: A national map is required to be included in the Figure1 to help explaining the spatial location of the study areas and why it is critically important to use this province as a case study. It is necessary to indicate the locations of meteorological stations in this figure to indicate if these locations could represent the terrain characteristics.

2.3

Line 96: “…multiple meteorological elements” References?

Results

Line 226: Have you ever indicated the actual demand at somewhere? If not, it is inappropriate to give this conclusion.

Line 232-243: What is the reason for the distribution of this trend? The land quality and use? You indicated that the cultivated land - population carrying capacity has more space to use. Does this mean that the relationships between land and population account for this trend?

This need to be discussed. 

Discussion

In this discussion, it mainly discussed the results for two of the research aims. The other two aims have been fully discussed. I recommended to divide this discussion into subsections according to the research aims and add more information for the discussion of The population carrying capacity index and Population-food relationship.

Line 295-297: This goes too far from this study. And I do not think it makes any sense to study the law of climate change in a province.

Line 299: The results should be compared and discussed to more literature.

Line 304-305: This recommendation is inappropriate. It is due to the limitations on available land for agriculture production and the environmental effects on increasing the supply side. So instead of expanding the production areas, it is better to increase the land use efficiency, land quality and security to promote the sustainable development of agriculture and improve the food security. To improve the land use efficiency and quality, more environmentally friendly technologies and digitalization approaches could be applied such as digital soil mapping and management strategies could be use for accurate and efficient agricultural management (e.g., irrigation, fertility and carbon sequestration) and yield improvement. Here are some references which could help you to revise and discuss the results.

McBratney, A. B., Santos, M. M., & Minasny, B. (2003). On digital soil mapping. Geoderma, 117(1-2), 3-52.

Zhao, D., Wang, J., Zhao, X., & Triantafilis, J. (2022). Clay content mapping and uncertainty estimation using weighted model averaging. Catena, 209, 105791.

Zhao, D., Arshad, M., Wang, J., & Triantafilis, J. (2021). Soil exchangeable cations estimation using Vis-NIR spectroscopy in different depths: Effects of multiple calibration models and spiking. Computers and Electronics in Agriculture, 182, 105990.

Zhao, D., Li, N., Zare, E., Wang, J., & Triantafilis, J. (2020). Mapping cation exchange capacity using a quasi-3d joint inversion of EM38 and EM31 data. Soil and Tillage Research, 200, 104618.

Zhao, D., Zhao, X., Khongnawang, T., Arshad, M., & Triantafilis, J. (2018). A Vis‐NIR Spectral Library to Predict Clay in Australian Cotton Growing Soil. Soil Science Society of America Journal, 82(6), 1347-1357.

Author Response

(The authors gave the same response as above.)

Reviewer 3 Report

I advise the authors to consider addressing the following comments:

  1. It should be ensured that words included in the title are not iterated as key-words.
  2. Lines 52-54: This sentence should be referred to and supported by literature.
  3. In section 2.1, more information should be added about the study area. For instance, employment and population of the area must be added. Finally, unit 2.1 should be supported by literature sources.
  4. Remove period (.) from line 293.
  5. Lines 304 – 306: The authors should explain where the increase in demand comes from. This sentence should also be referred to and supported by literature.
  6. Line 310: The word 'publicity' has been a bit off-centre and I think it should be removed.

Author Response

(The authors gave the same response as above.)

Round 2

Reviewer 2 Report

After reading through authors’ revisions, I believe this paper has been improved a lot. But there are still many issues authors should address. Authors should further do major revisions.

Major

* Line 59-70, please describe such information more scientific. Readers always want to read a scientific research gap, not general information. Please rewrite this paragraph to generate an ‘academic’ research gap.

* Fig.1, the national map does not include South China Sea, it is not correct. Please also include the unit of the DEM. I guess it should be m. Moreover, it is not clear to see the Gansu province in the China map, please use other colour to make it more recognisable.

* Previous Q9 ‘Line 226: Have you ever indicated the actual demand at somewhere? If not, it is inappropriate to give this conclusion’. Authors have just mentioned ‘we have modified’, where is it?

* Line 355-357, it is too superficial to just mention the digital soil map. Please link to my previous suggested literature to enhance the discussion.

Clay content mapping and uncertainty estimation using weighted model averaging. Catena, 209, 105791.

Soil exchangeable cations estimation using Vis-NIR spectroscopy in different depths: Effects of multiple calibration models and spiking. Computers and Electronics in Agriculture, 182, 105990.

Mapping cation exchange capacity using a quasi-3d joint inversion of EM38 and EM31 data. Soil and Tillage Research, 200, 104618.

A Vis‐NIR Spectral Library to Predict Clay in Australian Cotton Growing Soil. Soil Science Society of America Journal, 82(6), 1347-1357.

Minor:

* Line 28, p < 0.01, not P<0.01.

* Line 28, before ‘it shows’, there should be a dot to end this sentence, not comma.

* Line 28-30, this is a bad sentence, I believe it should be ‘It shows a trend of decreasing from south to north, and whole area undergoes a decreasing trend consistently, indicating that the population-food relationship in eastern Gansu tends to be balanced’

* Keywords, too long, please separate it into Cultivated land, population carrying capacity, Climate production potential. Otherwise, readers cannot find your work based on such keywords, no one use a six-word keyword.

* I suggest you use food security, but you used food safety in line 42, please remember ‘food security’ is now a fancy word.

* line 57, it is not good to use ‘this year’. If your paper is written in 2021, but published in 2022, so what do you mean by ‘this year’? Please just describe the specific year.

* Line 335, please do not describe others’ work through ‘:’. You just need to mention how their results supports/defends your results.

Author Response

dear reviewer
The article has been revised, please point out if there are any problems.
good luck.
